# Quantification of Zonisamide in Dried Blood Spots and Dried Plasma Spots by UPLC–MS/MS: Application to Clinical Practice

**DOI:** 10.3390/molecules27154899

**Published:** 2022-07-31

**Authors:** Milena Rmandić, Ana Stajić, Jasna Jančić, Janko Samardžić, Nebojša Jović, Anđelija Malenović

**Affiliations:** 1Department of Drug Analysis, Faculty of Pharmacy, University of Belgrade, Vojvode Stepe 450, 11000 Belgrade, Serbia; mrmandic@pharmacy.bg.ac.rs; 2AstraZeneca UK Limited Representative Office, Milutina Milankovića 1i, 11000 Belgrade, Serbia; astajic@pharmacy.bg.ac.rs; 3Clinic of Neurology and Psychiatry for Children and Youth, Faculty of Medicine, University of Belgrade, Dr Subotića 6a, 11000 Belgrade, Serbia; jasna.jancic.npk@gmail.com (J.J.); jankomedico@yahoo.es (J.S.); npk.zmaj@gmail.com (N.J.); 4Institute of Pharmacology, Clinical Pharmacology and Toxicology, Faculty of Medicine, University of Belgrade, Dr Subotića 1, 11000 Belgrade, Serbia

**Keywords:** zonisamide, dried plasma spots, dried blood spots, image processing, hematocrit, UHPLC–MS/MS method, clinical practice

## Abstract

In this research, a UHPLC–MS/MS method was developed and validated for the determination of zonisamide in dried plasma spots (DPS) and dried blood spots (DBS). Detection of zonisamide and internal standard, 1-(2,3-dichlorphenyl)piperazine, was carried out in ESI+ mode by monitoring two MRM transitions per analyte. Total run time, less than 2.5 min, was achieved using Acquity UPLC BEH Amide (2.1 × 100 mm, 1.7 µm particle size) column with mobile phase comprising acetonitrile–water (85:15%, *v*/*v*) with 0.075% formic acid. The flow rate was 0.225 mL/min, the column temperature was 30 °C and the injection volume was 3 µL. Desolvation temperature, desolvation gas flow rate, ion source temperature and cone gas flow rate were set by the IntelliStart software tool in combination with tuning. All of the Guthrie cards were scanned, and DPS/DBS areas were determined by the image processing tool. The influence of hematocrit values (20–60%) on accuracy and precision was evaluated to determine the range within which method for DBSs is free from Hct or dependency is within acceptable limits. The validated method was applied to the determination of zonisamide levels in DPS and DBS samples obtained from patients confirming its suitability for clinical application. Finally, the distribution of zonisamide into the red blood cells was estimated by correlating its DPS and DBS levels.

## 1. Introduction

Zonisamide (1,2-benzisoxazole-3-methanesulfonamide) is an antiepileptic drug (AED) chemically distinct from other AEDs, with a dual mechanism of action, and thus effective in patients whose seizures are resistant to other anticonvulsants. Its biotransformation products are neither active nor toxic since 20% of zonisamide undergoes acetylation to form N-acetyl zonisamide, and 50% is reduced to form the open ring metabolite 2-sulfamoylacetyl phenol (SMAP), which is eliminated as glucuronide, while 30% is excreted unchanged in the urine [1]. The drug is not highly bound to human plasma proteins, but its binding affinity for red blood cells (RBCs) is eight times higher than that for plasma proteins, and a marked concentration of zonisamide can be observed in human RBCs [2].

For the determination of zonisamide (ZNS) in plasma/serum HPLC [3,4], LC-MS [5], LC-MS/MS [6,7] and UHPLC–MS/MS [8] methods were reported, while in dried plasma spots (DPSs) it was quantified by HPLC-UV method [9]. The reported HPLC methods suffer from incorrect validation procedures and inadequate lower limit of quantification (LLOQ), while LC-MS methods encountered several disadvantages such as a lack of specificity due to the application of selected ion monitoring (SIM) rather than multiple reaction monitoring (MRM) modes [5], inadequate LLOQ [6], laborious sample preparation procedures [5,7] or compromised cost-effectiveness due to long analytical run [7]. UHPLC–MS/MS dramatically improved analytical run times without sacrificing peak resolution, but its clinical application is limited only to serum after protein precipitation for sample pre-treatment [8].

Thus far, a method suitable for quantification of ZNS in both plasma and blood was not proposed, providing means to correlate obtained concentrations and apply findings in routine clinical practice. Therefore, the aim of this research was to develop the first UPLC–MS/MS method for the quantification of ZNS in DPSs and dried blood spots (DBSs) with the aid of image processing for precise estimation of DBS and DPS areas after scanning of the Guthrie card on which the appropriate matrix volume was spotted. For the appropriate collection of DBSs, regardless of hematocrit (Hct) value, we applied a previously verified procedure for consistent application of a blood sample volume and formation of spots of the uniform area [10].

Furthermore, both methods were validated in accordance with Food and Drug Administration (FDA) [11], European Medicines Agency (EMA) [12] and International Council for Harmonisation of Technical Requirements for Pharmaceuticals for Human Use (ICH) M10 guidelines [13] on bioanalytical method validation, as well as European Bioanalysis Forum (EBF) recommendations on DBS assays [14]. The DBS-based method was validated using 5 Hct levels (20.2%, 30%, 42%, 50.2% and 60%) to determine the range within which the method is independent of Hct or dependency is within acceptable limits. Finally, the suitability of the developed methods for the clinical application was confirmed by analyzing samples obtained from epileptic children and youth.

## 2. Materials and Methods

### 2.1. Chemicals and Materials

Reference standard of ZNS, as well as the internal standard, 2,3-dichlorphenilpiperazine was obtained from LGC standards (LGC Group, Teddington, UK) (Figure 1). Acetonitrile, methanol and water, all LC-MS grade, were purchased from Thermo Fischer Scientific (Massachusetts, USA). Formic acid (purity 99%), acetic acid, ammonium formate and ammonium acetate, also LC-MS grade, were obtained from Sigma Aldrich (Taufkirchen, Germany). Vacutainer K2EDTA tubes for blood collection were supplied by Becton Dickinson, UK.

Air displacement Eppendorf pipette with variable volume 1–5 mL (Eppendorf AG, Hamburg, Germany) was utilized during the preparation of blood samples of target Hct values. Whatman 903 cellulose-based Guthrie cards utilized for DBS and DPS sample collection were procured from GE Healthcare (Buckinghamshire, UK). Thermo automatic pipettes with fix volume of 50 µL, variable volume of 10–100 µL and 0.1–10 µL (Thermo Fischer Scientific, Massachusetts, USA) and non-commercial puncher were used during the sample deposition onto the blotting paper, as well as during analytical sample preparation procedure. Fisherbrand™ Zipper Seal Sample Bags (Fisher Scientific, Hampton, VA, USA) with Sorb-It 1 g desiccant packs (Clariant, Ahrensburg, Germany) were used for card storage. Millipore membrane filters of 0.22 µm pore size and 47 mm diameter were purchased from Sigma Aldrich (Merck KGaA, Darmstadt, Germany). Blank plasma was obtained from Sanquin (Amsterdam, The Netherlands), while blank blood was obtained from healthy volunteers. Plasma and blood samples were collected from patients of the Clinic of Neurology and Psychiatry for Children and Youth in Belgrade, Serbia, who are using ZNS in therapy.

### 2.2. Instrumentation

The chromatographic analysis was performed on the Waters Acquity UPLC H-Class system. Appropriate chromatographic behavior of ZNS and IS for a total run time of less than 2.5 min was achieved using Acquity UPLC BEH Amide (2.1 × 100 mm, 1.7 µm particle size) column and isocratic elution. The mixture of acetonitrile and water (85:15%, *v*/*v*), with added formic acid to the final content of 0.075%, was selected as the optimal mobile phase composition. The flow rate was 0.225 mL min^−1^, the column temperature was maintained at 30 °C and the injection volume was 3 µL.

The analysis was conducted on Waters Xevo TQD, a triple quadrupole mass spectrometer armed with an electrospray ion source (ESI). Detection and quantification of the ZNS and IS were carried out in the positive electrospray ionization (ESI+) followed by scanning in multiple reaction monitoring (MRM) mode. Two MRM transitions were used/monitored for each analyte. Waters Xevo TQD’s operating parameters such as desolvation temperature, desolvation gas flow rate, ion source temperature and cone gas flow rate were set at 500 °C, 1000 L h^−1^, 150 °C and 10 L h^−1^, respectively, while the optimal value of collision energy (CE), cone voltage (CV) and capillary voltages for each m/z transition was set by IntelliStart software tool in combination with tuning. Characteristic m/z transitions (parent > product ion), as well as CE and CV for each transition, are listed in Table 1.

EBA 20 Hettich centrifuge (Hettich GmbH & Co. KG, Tuttlingen, Germany) and Mythic 18 hematology analyzer (Orphée Medical, Geneva, Switzerland) were utilized during the preparation of blood samples of target Hct levels. Vortex Genie 2 Digital (Scientific Industries, Inc., Bohemia, NY, USA) was employed during the sample preparation procedure. All DPS and DBS samples were scanned by HP Scanjet 4070 Photosmart Scanner (HP, Palo Alto, CA, USA).

### 2.3. Software

Mass Lynx V4.1 software (Waters Corporation, Milford, MA, USA) provided the execution of all operations at the Waters Acquity UPLC/Xevo TQD systems. The IBM^®^ SPSS^®^ Statistics 20 (IBM Corporation, Armonk, New York, NY, USA) and Microsoft^®^ Office Excel 2013 software packages (Microsoft Corporation, Redmond, Washington, DC, USA) were used to process the results and perform the necessary statistical tests. The image processing tool of Matlab R2018b software (MathWorks, Natick, MA, USA) was employed to determine the accurate plasma and blood spot area. The Greenness Analytical calculator (AGREE tool) was used to calculate the green score and evaluate ecologic impact, i.e., the greenness of the developed and validated UHPLC–MS/MS method [15].

### 2.4. Sample Preparation

#### 2.4.1. Preparation of Calibration Standards and Quality Control Samples

The stock solution of zonisamide (ZNS) was prepared by dissolving an appropriate quantity of reference standards of ZNS into 0.9% sodium chloride solution (0.9% NaCl) to obtain a concentration of 2000 µg mL^−1^, while a stock solution of internal standard (IS) was prepared by dissolving reference standard of 2,3-dichlorphenilpiperazine into acetonitrile to a concentration of 100 µg mL^−1^. Until further use, prepared stock solutions were stored at 2 to 8 °C. Stability of the ZNS and IS stock solution, under specified conditions, was experimentally verified for the period of six and two months, respectively (Section 3.3.7).

The stock solution of IS was further diluted in the mobile phase to the concentration of 20 ng mL^−1^. Working solutions of ZNS were prepared by diluting stock solution with 0.9% NaCl to final concentrations of 5, 10, 20, 40, 100, 200, 400 and 1000 µg mL^−1^ for DPS and 2.5, 5, 10, 20, 50, 100, 200 and 500 µg mL^−1^ for DBS. In the next step, drug-free matrices (plasma and blood) obtained from the health volunteers were mixed with a proper working solution at a 19:1 ratio providing eight non-zero calibration standards for plasma, i.e., for blood. Final concentrations of the plasma calibration standards were in the range 0.25–50 µg mL^−1^ (0.25, 0.5, 1, 2, 5, 10, 20 and 50 µg mL^−1^), while the final concentrations of the blood calibration standards were in the range 0.125–25 µg mL^−1^ (0.125, 0.25, 0.5, 1, 2.5, 5, 10 and 25 µg mL^−1^).

An additional stock solution of ZNS (2000 µg mL^−1^) was prepared by dissolving separately weighted ZNS reference standard in 0.9% NaCl. After diluting the appropriate volume of this stock solution with 0.9% NaCl, quality control (QC) working solutions were prepared in four concentration levels: 5, 15, 150 and 760 µg mL^−1^ for DPS and 2.5, 7.5, 75 and 380 µg mL^−1^ for DBS. Finally, QC samples at concentrations of 0.25 µg mL^−1^ (LQCp), 0.75 µg mL^−1^ (QC1p), 7.5 µg mL^−1^ (QC2p) and 38 µg mL^−1^ (QC3p) for DPS and 0.125 µg mL^−1^ (LQCb), 0.375 µg mL^−1^ (QC1b), 3.75 µg mL^−1^ (QC2b) and 19 µg mL^−1^ (QC3b) for DBS were obtained by twenty-fold dilution of each QC working solution in drug-free plasma and blood. Calibration samples for the DBS were prepared in the blood of Hct 42%, which was considered a reference/nominal level. In addition to the reference Hct level (42%), the QC samples for DBSs were also prepared with blood in four additional levels of Hct: 20.2%, 30.0%, 50.2% and 60.0%.

#### 2.4.2. Preparation of Blood Samples of Target/Desired Hematocrit Values

Whole blood was drawn from a healthy female volunteer by venipuncture. After collection, the Hct level of collected blood was determined by a hematology analyzer. Based on the determined Hct level, accurate plasma volumes to be removed or added during the preparation of desired Hct levels (20%, 30%, 40%, 50%, 60%) were calculated. Additionally, blood was centrifuged at 6000 rpm for 5 min in order to separate plasma from blood cells. The previously calculated volume of plasma was added or removed from the samples, and after gentle mixing, the Hct of each sample was measured to verify/check the Hct level. The obtained final Hct levels were 20.2%, 30.0%, 42.0%, 50.2% and 60.0%.

#### 2.4.3. Preparation of Dried Plasma Spot and Dried Blood Spot

Thirty microliters of plasma samples (blank plasma, calibration and QC plasma samples and patients’ plasma) and fifty microliters of blood samples (blank blood, calibration, QC blood samples and patients’ blood) were spotted in four replications on the Whatman 903™ card. The influence of different matrix volumes (plasma and blood) spotted on the card on ZNS quantification was investigated by depositing 10 µL and 20 µL of plasma QC samples and 10 µL, 20 µL and 30 µL of blood QC samples on the blotting paper. After spotting, cards were let to dry at room temperature for 24 h, scanned, packed in zipper seal bags with desiccant and stored in a refrigerator.

#### 2.4.4. Analytical Sample Preparation

From the center of the dried matrix spot (DMS that implies both DBS and DMS), a 5 mm punch was taken by the non-commercial puncher and replaced with the 2 mL safe-lock Eppendorf tube. One hundred μL of extraction solution (20 ng mL^−1^ IS solution in the mobile phase) was added to each Eppendorf tube. The blank matrix samples, analyzed during the analytical run for the calibration curves creation, were prepared by the addition of the mobile phase solely to the 5 mm punch taken from the center of the blank DMS. Then, the tubes were vortexed for 15 min at 680 rpm. The solution with extracted components was then transferred from each Eppendorf tube to the pre-marked insert vials.

### 2.5. Dried Plasma Spot and Dried Blood Spot Scanning and Spot Area Determination Procedure

Prepared DMS samples were scanned in the grayscale mode, and the obtained scans were saved in tif format. Together with each DMS sample, a solid colored square of 1 cm2 area was scanned as a reference object. Scans were further processed by applying the algorithm developed by image processing Toolbox^TM^ in the Matlab software. In such a manner, image data were converted into the areas. The areas automatically expressed as cm^2^ are obtained by the following Equation (1):(1)$$\begin{array}{ll} DMS\;area\;in\;cm^{2} & \\ & \;=\frac{DMS\;pixel\;count}{Reference\;object\;pixel\;count}\\ & \;\times Area\;of\;reference\;object\;in\;cm^{2} \end{array}$$

The applied image processing approach is described in detail in our previously published paper [8].

### 2.6. Patients’ Sample Collection

For the analysis, overall, 23 DBS and DPS samples were collected from the patients according to the approval of the Ethics Committee of the Clinic of Neurology and Psychiatry for Children and Youth in Belgrade (approval number 1-98/2, 9 January 2017). Patients included in the study were from 13 months to 23 years old. The daily dose taken by each patient varied from 50 mg to 400 mg of ZNS. All but one patient were in a steady state. From the hospitalized patients, 3 mL of blood were collected at four time points (2 h, 4 h and 8 h after the dosing and 30 min before the next dosing), while in the case of the ambulant patients, the same volume of blood was taken once, mostly 2 h after taking a dose. Immediately after blood collection, the Hct value was determined, and then 50 µL of blood was applied onto the Guthrie card in four replicates. Centrifugation of remaining blood to separate plasma from the blood cells was followed by deposition of 30 µL of plasma onto the blotting paper in four replicates. Guthrie cards were left to dry at room temperature for 24 h, scanned, packed in zipper seal bags with desiccant and stored in a refrigerator.

## 3. Results and Discussion

For the quantification of ZNS in DMSs by UPLC–MS/MS and consequent application to clinical practice, a single 5 mm diameter disc was planned to be taken from the center of the DBS/DPS. Moreover, the expected amount of ZNS in approximately 8–9 µL of blood and 4.5 µL of plasma would be apparently low. In an effort to provide a reliable determination of ZNS present in DBS and DPS in limited quantities, the maximization of ZNS signal intensity and reproducibility were imposed as a crucial aim of the method development phase. Additionally, sample collection was planned and organized in line with the literature review findings that ZNS undergoes rapid and complete absorption achieving peak plasma concentrations 2 to 4 h with steady-state plasma concentrations ranging from 1.9 to 55.3 µg mL^−1^ after 10 to 12 days of dosing [2].

### 3.1. MS and MRM Method Development

Initially, an adequate ionization technique was selected by examining the potential of electro spray ionization (ESI) and atmospheric pressure chemical ionization (APCI), both employed/operated in the positive and negative ion mode, to contribute to the intensity and stability of the ZNS signal. A standard solution of ZNS, in a concentration of 1 µg mL^−1^, was prepared by dissolving the reference standard into the acetonitrile–water mixture (50/50, *v*/*v*) and directly infused into the mass spectrometer. In reference to ZNS MS response, ESI + ionization mode was selected as more appropriate. In this phase of UPLC–MS/MS method development, the MS parameters (tune’s parameters) such as capillary voltage, source temperature, desolvation temperature, desolvation gas flow and cone gas flow were set as follows: 3.8 kV, 150 °C, 500 °C, 650 L h^−1^ and 10 L h^−1^, respectively. Furthermore, MRM method development for ZNS as well as for the IS was completed with the aid of the IntelliStart tool of MassLynx software. Two stable transitions of single charged molecular ion [M + H]^+^ as the precursor ion to product ions were found for both ZNS and IS. Quantification was based on the transition with the most intensive/abundant product ion for each analyte, while the other transition was used for confirmation. Characteristic transitions for ZNS and IS, with corresponding cone voltage and collision energy for each of them, are shown in Table 1. The dwell time during the MRM scanning was set at 0.100 s. With the development of the tune method and MRM method, preconditions for subsequent optimization of the chromatographic method were established.

### 3.2. UHPLC Method Development

During the preliminary testing, the influence of the stationary phases’ chemistry, mobile composition, i.e., type of the organic modifier (acetonitrile and methanol) and mobile phase additives (formic acid and ammonium formate) on the chromatographic behavior and method sensitivity, such as ZNS signal intensity, were investigated. Under the observed experimental conditions, the Acquity UPLC BEH Amide column was selected as the most adequate from the aspects of the retention, separation and ZNS’s peak shape. Both organic modifiers enabled the adequate separation of the ZNS and IS, but acetonitrile provided higher sensitivity than methanol. Sensitivity was also further improved by the addition of formic acid to the mobile phase in a content higher than 0.05%. As a result of the preliminary phase, appropriate separation of ZNS and IS for the total run time of less than 2.5 min was achieved using Acquity UPLC BEH Amide (2.1 × 100 mm, 1.7 µm particle size) column with isocratic elution. The investigated factors (acetonitrile content from 80–90%, formic acid content from 0.05–0.1%, as well as column temperature from 25–40 °C) were in agreement and with a positive effect on ZNS signal and retention in the investigated experimental domain. The optimal mobile phase composition was defined as a mixture of acetonitrile and water (85:15%, *v*/*v*), with added formic acid to the final content of 0.075%, with a flow rate of 0.225 mL min^−1^, column temperature maintained at 30 °C and injection volume of 3 µL. Eventually, at the final stage of UPLC–MS/MS method development, we examined whether the established chromatographic conditions jeopardize the performance of the previously created MS/MS method considering the intensity and reproducibility of ZNS signal in reference standard solution with the concentration of 100 ng mL^−1^. It was found that the only crucial alteration was in the desolvation gas flow rate, which was set at 1000 L h^−1^ to accommodate the mobile phase flow rate.

### 3.3. UHPLC–MS/MS Method Validation

The following sub-headings describe the validation of the developed UPLC–MS/MS method, conducted according to the current FDA [9], EMA [10] and ICH M10 guidelines [11] on bioanalytical method validation with respect to the next aspects: selectivity (endogenous and cross-analyte), LLOQ, calibration curve, accuracy, precision [9,10,11], carry-over (instrumentation and spot to spot), matrix effect [10,11] and extraction recovery [9], dilution integrity [9,10,11] and incurred sample analysis [9,10,11]. Additionally, matrix spot volume homogeneity and matrix spot homogeneity, as well as the influence of various Hct values on accuracy and precision, were evaluated in accordance with EBF recommendations on DBS assays [12]. By striving to gain insight into the ability of the method to reliably quantify ZNS in DBS samples obtained from patients with Hct values different than normal level (Hct around 40%), the scope of certain tests performed during validation was expanded. Accordingly, in addition to DBS calibration and QC samples with Hct 42%, the DBS calibration samples and QC samples with Hct values of 20.2%, 30.0%, 50.2% and 60.0% were considered during the calibration curves creation as well as during the evaluation of accuracy and precision. The results of these assays directed the rest of the validation and provided the information relevant for the routine application of the method. Results of this additional evaluation, as well as protocol characteristics themselves, are described in detail in the Section 3.3.10.

#### 3.3.1. Selectivity

With the aim to find whether the proposed method is able to differentiate the ZNS and IS from the interferences caused by endogenous substances as well as co-administrated drugs, blank plasma and blood obtained from the six healthy individuals were tested. As a result of the insight into the patients’ health records, the following co-administrated drugs were considered as potential interferences: valproic acid, levetiracetam, phenytoin, melatonin, carbamazepine, acetaminophen and ibuprofen. Before applying to the Guthrie cards, each of these analytes was spiked into the blood, as well as plasma samples up to the concentration of 10 µg mL^−1^. The DMS samples thus formed were further processed and analyzed. According to the requirements of current guidelines, responses detected and attributable to interfering components should not be more than 20% of the analyte response at the LLOQ and not more than 5% of the IS response in the LLOQ sample for each matrix [11,12,13]. Obtained MRM chromatograms indicated that there were no detectable responses/interference at the retention times of ZNS as well as IS. The MRM chromatogram of the blank matrices, as well as the MRM chromatogram of IS and ZNS at LLOQ levels for the plasma and blood, are presented in Figure 2.

#### 3.3.2. Calibration Curves for ZNS Quantification in DPS and DBS

Four analytical runs were realized, each one containing one calibration curve with eight non-zero calibration standards in the range 0.25–50 µg mL^−1^ for DPS and 0.125–25 µg mL^−1^ for DBS. These ranges covered the expected ZNS levels when the drug is administered in the twice-daily dosing regimen. Likewise, calibration curves were created by analyzing DBS calibration standards with Hct of 42%; the calibration curves for the DBS calibration standards with Hct values of 20.2%, 30.0%, 50.2% and 60.0% were also set. A linear regression model (*y* = *ax* + *b*) was utilized to establish the relationship between ZNS’s concentration (*x*) and the peak area ratio of ZNS and IS (y). Curve weighting was carried out in accordance with the least sum of percent relative error (%RE) per linear regression. As the most suitable, 1/x^2^ weighting factor was selected. Back calculated concentrations of the calibration standards for each analytical run complied with acceptance criteria (±15%, except for the LLOQ standard where the allowed deviation was ±20% of the nominal value) [11,12,13]. Additionally, each established relationship was accompanied by a correlation coefficient (r) greater than 0.9900. The slope (*a*) and intercept (*b*) of the linear weighted models for each analytical run, the correlation coefficients and the %REs displayed for the largest deviation of the back-calculated concentration from the nominal concentration are shown in Table 2. Certain regression models related to the DBS with Hct values different from the nominal Hct value were included in the accuracy and precision testing (Section 3.3.10).

#### 3.3.3. Carry-Over

No instrumentation carry-over was observed when the blank solvent mixture was injected after the injection of the ZNS calibration standards of the highest concentration for both matrices. Moreover, spot-to-spot carry-over (carry-over from the mechanical punching) was not observed when the processed punches of the blank matrices were obtained after punching from the DBS and DPS with the highest concentration of the ZNS were injected.

#### 3.3.4. Accuracy and Precision

Within-run accuracy and precision were evaluated simultaneously by analyzing five samples per each of four concentration levels (LLOQ, low QC, medium QC, high QC) in a single analytical run performed in triplicate. Between-run accuracy and precision were estimated by analyzing results obtained from the three single runs over two days for both matrices. Accuracy was reported as a percentage deviation of calculated concentration from the nominal concentration (bias,%) and obtained results were in accordance with the guidance requirements (mean value should be within ±15% of the nominal values for each QC sample, except for the LLOQ which should be within ±20% of the nominal value) [9,10,11]. Precision results were expressed as a % Relative Standard Deviation (%RSD) for each level. Calculated %RSD values attained acceptance criteria (RSD value should not exceed 15% for the QC samples, except for the LLOQ, which should not exceed 20%). Results for within-run and between-run accuracy and precision for the DPS and DBS (Hct 42%) samples are listed in Table 3.

#### 3.3.5. Matrix Effect

Matrix effect (ME) [12,13] was tested using two sets of samples-so-called matrix-present and matrix-absent samples, that were prepared in triplicate in three concentration levels (low QC, medium QC and high QC) for each of two batches of plasma and blood (Hct 42%). Certain volumes of the ZNS working solutions prepared into the 0.9% sodium chloride at concentrations corresponding to the QC concentrations for plasma and blood were spotted onto the pre-punched discs obtained from the empty Guthrie cards. The volumes of added working solutions represented the volume available in the one punched disc of the DMS (*V_opd_*). The volumes *V_opd_* were calculated from the following proportion:(2)$$V_{opd}=\frac{V_{m}}{P_{dms}}\;P_{pd}$$
where *V_m_* corresponds to the volume of matrix spotted onto the cards, which in this study was 30 µL for the plasma and 50 µL for the blood; *P_dms_* indicates the area of certain dried matrix spot; and *P_pd_* is the area of the disc punched from the DMS. Each disc obtained by an uncommercial puncher has a diameter of 5 mm; consequently, *P_pd_* has a constant value of 0.1963 cm^2^. While *V_m_* and *P_pd_* are known values, the value of the *P_dms_* should be determined. When striving to reliably determine *P_dms_*, the Image Processing Toolbox in Matlab software was employed. According to Equation (2), one disc punched from the DPS contained approximately 3.6 µL of plasma and approximately 9.2 µL of blood (Hct 42%). Adequate volumes, based on the calculation of the ZNS’ working solutions of certain concentrations, were applied to the pre-punched discs. Furthermore, extraction of the ZNS in the case of the matrix present samples was conducted with 100 µL of blank matrix (blood and plasma) extract, which contained IS in a concentration of 20 ng mL^−1^, while the ZNS for the matrix-absent samples was extracted with 100 µL of the extraction solution. According to the recommendations of the EMA guidelines, ZNS, as well as IS matrix factor (MF), were calculated by dividing the peak area of each analyte obtained for the matrix-present sample by the peak area obtained from the matrix-absence sample and expressed as a percentage value [10]. Additionally, by dividing the MF of the analyte by the MF of the IS, the IS normalized MF was calculated [12]. Since ICH M10 guidelines give a recommendation that the accuracy should be within ±15% of the nominal concentration and the precision expressed as %RSD should not be greater than 15% in all individual matrix batches; the % bias as the accuracy parameter and%RSD as the precision parameters were calculated for each QC level for both sets of samples. The average value of the MFs (MF of the ZNS, MF of the IS and IS normalized MF) obtained for the two batches of both matrices at each QC level are presented in Table 4.

The obtained results confirmed the absence of a significant matrix effect. Additionally, Table 4 contains the CZNS bias% data obtained by dividing the deviation of the ZNS concentration obtained in matrix-present from ZNS concentration obtained in matrix-absent samples, with ZNS determined in matrix-absent samples. All calculated values of the CZNS bias% were within ±3.6%, which confirmed that in this study, the effects of the matrix on ZNS quantification were negligible.

#### 3.3.6. Recovery

Recovery testing was conducted with two batches of samples in three QC levels with three replicates per level [11]. Both matrices, plasma and blood (Hct 42%), were employed during the testing. The first set consisted of DMSs obtained by spotting appropriate QC samples onto the Guthrie card. The second set contained DMSs obtained by deposition of blank matrices onto the blotting cards. Extraction of both sets of DMSs was performed as described in the Section 2.4.4. Additionally, extracts of the second set (blank plasma and blood) were spiked post-extraction with an adequate volume of ZNS working solution in concentrations corresponding to QC levels. The volumes of added standard solutions corresponded to the volume available in the one disc punched from the DMS, which, based on the previous calculation, was 3.6 µL of plasma and approximately 9.2 µL of blood (Hct 42%). After analyzing the prepared samples, Recovery was evaluated by comparing the analytical results of extracted samples with corresponding extracts of blank plasma and blood spiked with the ZNS post-extraction. Obtained results expressed as a percent were reported in Table 5.

#### 3.3.7. Stability Testing

Stability testing encompasses evaluation of ZNS and IS stock and working solution stability testing, short and long-term stability of the DBS and DPS, as well as auto sampler stability of processed samples. The stability of ZNS and IS stock solutions stored at 2–8 °C was evaluated for the period of 6 and 2 months, respectively. Stock solutions’ dilution, with a mixture of acetonitrile/water (85/15%, *v*/*v*) to a concentration of 10 µg mL^−1^ for ZNS and 0.1 µg mL^−1^ for IS, preceded analysis. The aged solutions were analyzed against the freshly prepared ones. Stability of ZNS and IS working solutions were evaluated for a period of 7 days. Short term stability of ZNS in DMS was investigated at the two QC levels (low QC and high QC) for both matrices in triplicate. Estimation was performed by comparing the concentration of ZNS determined in DMSs stored before processing, 8 h at room temperature, with ZNS concentration determined in freshly prepared DMSs. Long-term stability was evaluated in triplicate at the low and high QC samples of both matrices stored, before analyzing, in a refrigerator at 2–8 °C for the period of six months. Processed aged samples were analyzed against the fresh DMS samples. The auto sampler stability was estimated for the processed DMS samples (low QC and high QC) placed in auto sampler for a 4.5 h period. Time was defined according to the maximal capacity of the auto sampler (two racks, each with a capacity of 48 vials) and the duration of the analytical run (2.5 min). The ZNS quantification was performed at the beginning of the sequence as well as 4.5 h later at the end of the sequence. The quantification results obtained for each QC sample at the beginning and after 4.5 h were compared. The results were obtained from the stability testing accomplished acceptance criteria, which prescribes that the mean concentration at each QC level should be within ±15% of the nominal concentration. Additionally, mutual comparison of concentrations determined at each QC level, before and after defined conditions, demonstrated deviation within ±15%. The outcomes of the stability testing (short-term stability, long-term stability and auto sampler stability of processed samples) are provided in Appendix A.

#### 3.3.8. Dilution Integrity

Testing included DMSs containing ZNS in excess concentrations of 50 µg mL^−1^ and 80 µg mL^−1^ for DBS and DPS, respectively. Through the processing phase, the primary samples were diluted with the extract of the blank matrices (blood and plasma) two and four times to the final concentrations of 25 µg mL^−1^ (dQC1b) and 12.5 µg mL^−1^ (dQC2b) for DBS and 40 µg mL^−1^ (dQC1p) and 20 µg mL^−1^ (dQC2p) for DPS. Five samples per dilution were analyzed. The concentration of ZNS in diluted DBS samples was determined with associated accuracy and precision: ±13.21% and 2.63% (RSD) for dQC1b and ±10.82% and 2% (RSD) for dQC2b. Accuracy and precision for determination in diluted DPS samples were: ±10.29% and 11.13% (RSD) for dQC1p and ±11.51% and 4.52% (RSD) for dQC2p. Obtained results accomplished the acceptance criteria, which prescribe that the mean accuracy of the analytical results obtained for the diluted QC should be within ±15% of the nominal concentration and the precision (% RSD) should not exceed 15% [11,12,13].

#### 3.3.9. Homogeneity (Matrix Spot Homogeneity and Matrix Spot Volume Homogeneity)

As part of the matrix spot homogeneity testing, punches from the different regions of the spot beyond the center were taken out. The assessment covered three QC levels for both matrices. According to the spot size, four punches were obtained from the DPSs, while from the DBSs, three punches were acquired. The five replications per region were made. Regardless of the region where the punch was made, the deviation of quantified ZNS concentrations was within ±15% of the nominal concentration, and the RSD calculated for each QC level did not exceed 15%. Obtained results implied that ZNS is uniformly distributed through the DMS.

In order to evaluate how the volume of the spotted blood and plasma affected the reliability of ZNS quantification, different volumes of the QC samples (low QC, medium QC and high QC) were applied to the cards in triplicate. The testing was conducted with 40, 30 and 20 µL of blood QC samples and with 20 and 10 of plasma QC samples. The ZNS concentrations determined in analyzed DBS and DPS samples were within ±15% of the nominal QC concentration, with RSD per each QC level below 15%. According to the obtained results, it was established that the deposition of blood and plasma samples in the assayed volume span did not jeopardize the reliability of the ZNS quantification.

#### 3.3.10. Additional/Miscellaneous Tests

This part of the study was focused on unveiling whether and how Hct affects the reliability of the ZNS quantification in DBS. In this regard, accuracy and precision testing were performed in the single run for QC samples with Hct different from the nominal Hct of 42%. The examination encompassed QC samples of Hct values of 20.2%, 30.0%, 50.2% and 60.0%, in addition to the QC sample with nominal Hct value. For the testing purpose, a calibration curve formed by analyzing DBS calibration standards with Hct of 42% and calibration standards with Hct values of 20.2%, 30.0%, 50.2% and 60.0% were computed. The important parameters of linear weighed regression models for each calibration curve are shown in Table 6.

With this approach, the peak area ratios obtained through the analytical runs were converted to ZNS concentrations by simultaneous application of regression models derived for the calibration standards with appropriate Hct and nominal Hct. Consequently, two sets of concentrations were obtained: the first set contained concentrations calculated regarding the regression model for the certain Hct, and the second set encompassed the concentrations calculated by applying the regression model for the nominal Hct. Important accuracy and precision parameters (%bias and% RSD) were calculated for each set of data and displayed in Table 6. Graphical relationship between%bias of calculated average concentration and nominal concentration ((calculated average concentration– nominal QC concentration)/nominal QC concentration) × 100) at each QC level and Hct value for the first and second set of data was illustrated in Figure 3.

These results obtained for the second set enabled monitoring of eventual change in accuracy and precision parameters with change in Hct value. In addition, obtained average values of the concentrations for each QC level of two sets of data were compared and reported as a percentage deviation of the concentration of the second set from the concentration of the first set (Table 6). Estimated parameters in both sets accomplished acceptance criteria for the accuracy and precision defined by official guidelines. Furthermore, the results acquired in this study indicated the absence of a significant difference in quantified ZNS concentrations among both sets of data. It was demonstrated that the method’s quantitative performances would not be compromised by fitting analytical run results into the calibration curve computed with DBS calibration standards of Hct value that did not match the Hct value of the patient’s DBS sample. The outcome of performed additional observations testified to the reliable quantification of the ZNS concentration regardless of its particular Hct if it falls within the evaluated region of Hct values. Therefore the analysis of patients’ samples can be carried out by fitting the results into the calibration curve formed with DBS calibration standards of Hct 42%, which was declared nominal in our study.

#### 3.3.11. Incurred Sample Reanalysis

All patients’ samples were reanalyzed seven days after the initial analysis. The results of the repeated analysis were compared with the initially obtained, according to the following equation:(3)$$\%\;difference=\frac{\left(repeat\;value-initial\;value\right)}{mean\;value}\times 100$$

According to the official guide, the percent difference should not be greater than 20% of their mean for at least 67% of the repeats [11,12,13]. All of the reanalyzed samples accomplished these acceptance criteria since the biggest deviations obtained for reanalyzed patients’ DMSs were −18.54 and +18.28 for DBS and DPS, respectively.

### 3.4. Greenness of Developed UHPLC–MS/MS Method

Due to awareness that all chromatographic methods suffer from environmental impact, after validation, our method was evaluated from the aspect of its greenness. The assessment was conducted according to the 12 principles of the Green Analytical Chemistry by using the Greenness Analytical calculator (AGREE tool) [15]. The characteristics of the method important for the evaluation and calculation of the overall green score were: at line simple, manual, but miniaturized extraction procedure and sample analysis, sample preparation procedure comprising less than three distinct steps, a maximal sample volume of 0.05 mL, analysis without derivatization agents, overall waste volume less than 0.6 mL, the analysis of one analyte in a single run and 24 samples per one hour, amount of toxic solvents is less than 0.5 mL and no threats to the operators were recognized. Utilization of LC-MS did not compromise greenness due to energy consumption, but the source of reagents appeared red since none of the reagents were from the bio base. The high value of the overall score (0.75 from maximal 1) indicated that the method has a low negative environmental impact, i.e., the method is green. The analytical greenness report with a graphical presentation of the assessment is provided in Appendix A.

### 3.5. Analysis of Patients’ DMSs

In succession to the validation, samples obtained from epileptic children and youth were analyzed in order to confirm the suitability of the developed UPLC–MS/MS method for clinical application. The determined Hct values of collected patients’ blood samples were from 31.0% to 47.0%. DBSs and DPSs were processed as described in Section 2.4.4. Before analysis of these samples, method performance was verified once again, and the acceptability of the analytical run was ensured by analyzing calibration standards and QC samples for both matrices, DPS as well as DBS (Hct 42%). The DBSs with initially quantified ZNS in the concentration above the upper limit of quantification (ULOQ) defined during the method validation were diluted in accordance with the previously evaluated dilution integrity (Section 3.3.8) and then reanalyzed, and ZNS concentrations were calculated indirectly, taking into account dilution factor of two or four. Concentrations of ZNS, quantified during the analysis of each DPS and DBS, as well as determined ZNS DBS to DPS ratio, are displayed in Table 7.

DBS:DPS ratio obtained for the analyzed samples was within the range 1.07–4.24, with a median of 1.70 and an interquartile range of 1.86. These results are in agreement with the fact that ZNS possesses a binding affinity to the intracellular compartment of RBCs [2,16] and literature available data reporting ZNS blood to plasma ratios of 2.7 ± 0.8 [17] and 2.63 [18]. ZNS binding to RBCs is non-linear due to a combined saturable binding to erythrocyte carbonic anhydrase at concentrations below 3 µg mL^−1^ and non-saturable binding (passive diffusion) at higher concentrations. As a consequence of the above-mentioned erythrocytes binding manner, an RBCs/plasma ratio of about 15 at low concentrations and about 3 at higher concentrations may occur [16]. Consequently, higher DBS:DPS ratios ranging from 3.11 to 4.24 in our study for samples with determined lower ZNS concentrations relative to other analyzed samples are not unexpected. Graphical presentation of the determined ZNS’ blood to plasma ratios regarding the Hct value and ZNS daily dose is illustrated in Figure 4.

The lowest ZNS concentrations were observed in sample 11, corresponding to the hospitalized patient that was not in a steady state, but on the initiation of therapy, individualization of the dosage regimen, with regard to the response. To this patient, 13-month-old ZNS was included in therapy ten days before sampling at a daily dose of 20 mg, but five days before sampling, the daily dose was increased to 40 mg.

## 4. Conclusions

For the determination of zonisamide (ZNS) in dried plasma spots (DPS) and dried blood spots (DBS), the UHPLC–MS/MS method, in ESI+ mode and monitoring two MRM transitions, was developed and validated. All of the Guthrie cards were scanned prior to further processing in order to estimate DBS and DPS areas by an algorithm developed by Image processing ToolboxTM in the Matlab software. The method was validated according to the current FDA, EMA and ICH M10 guidelines on bioanalytical method validation. Endogenous and cross-analyte selectivity of the developed method was confirmed, calibration curves were computed and accuracy and precision were estimated. All the required parameters were calculated, demonstrating adequate selectivity, linearity, accuracy and precision over the investigated range. Furthermore, experimentally was confirmed that instrumentation and spot-to-spot carry-over were not present and negligible effects of the matrix on ZNS quantification as well, while consistent and reproducible recovery values indicated the efficiency and reproducibility of the proposed extraction procedure. Dilution integrity was verified by testing accuracy and precision for determination in diluted DPS samples, and during incurred sample analysis, all of the reanalyzed samples accomplished established acceptance criteria. Matrix spot homogeneity testing suggested that ZNS is uniformly distributed through the DMSs, and matrix spot volume homogeneity assaying affirmed the reliability of ZNS quantification by spotting blood and plasma samples in the investigated volumes.

Examination of various hematocrit values (20–60%) influences accuracy and precision, implying that the analysis of patients’ samples can be carried out by fitting the results into the calibration curve formed with DBS calibration standards of Hct of 42%. The validated method was applied to determine ZNS in DPS and DBS samples obtained from patients treated with ZNS. The suitability of the developed method for clinical application was established since the concentration of ZNS was successfully determined both in samples of patients in a steady state and in a sample of the patient on the initiation of therapy. Equally important, DBS to DPS ratios of ZNS were determined, demonstrating the highest value for the sample with the lowest ZNS concentration due to their non-linear binding to the RBCs.

The newly introduced UPLC–MS/MS method for the determination of ZNS in DPSs and DBSs, compared to previously published, was cost-effective and eco-friendly, fully validated and with an adequate lower limit of quantification (LLOQ). Sampling methods such as DBSs and DPSs enabled straightforward sample preparation using mobile phase for direct dissolution of ZNS from a single 5 mm diameter disc, taken from the center of the DBS/DPS. Keeping in mind ZNS’s marked binding to RBCs, the suitability of this method for quantification of ZNS in both plasma and blood was of great importance since it provided means to correlate obtained concentrations and apply findings in routine clinical practice.

## Figures and Tables

**Figure 1 molecules-27-04899-f001:**
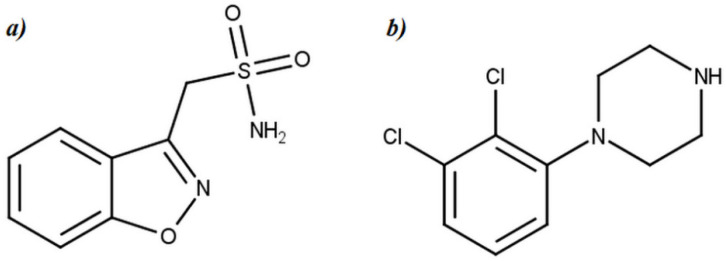
Chemical structure of (**a**) zonisamide and (**b**) internal standard (1-(2,3-dichlorphenyl)piperazine).

**Figure 2 molecules-27-04899-f002:**
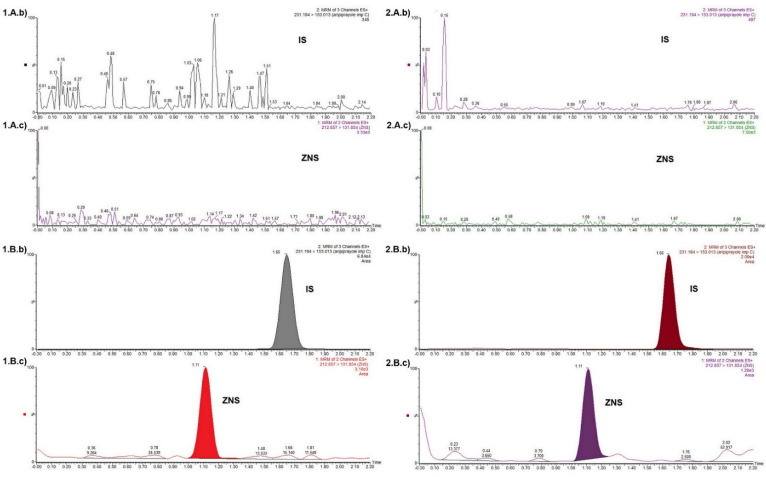
MRM chromatogram of the (**A**). blank matrices, as well as chromatogram of (**b**) IS and (**c**) ZNS at (**B**). LLOQ levels for the 1. DPS and 2. DBS.

**Figure 3 molecules-27-04899-f003:**
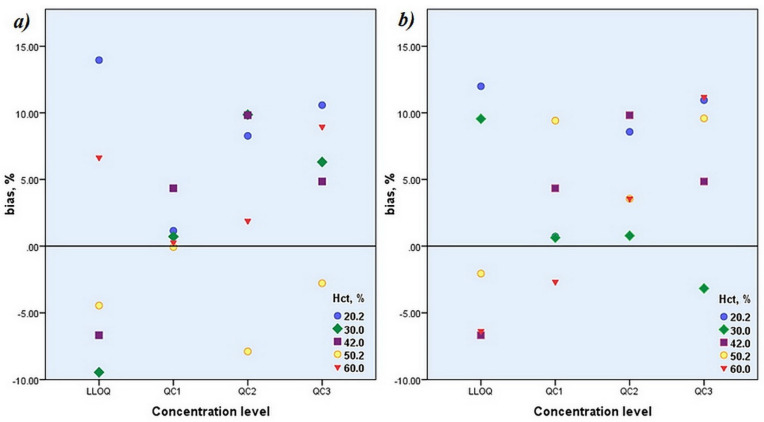
Relationship between% bias of calculated average concentration and nominal concentration-concentration calculated by applying regression model derived for the calibration standards with (**a**) appropriate Hct and (**b**) nominal Hct.

**Figure 4 molecules-27-04899-f004:**
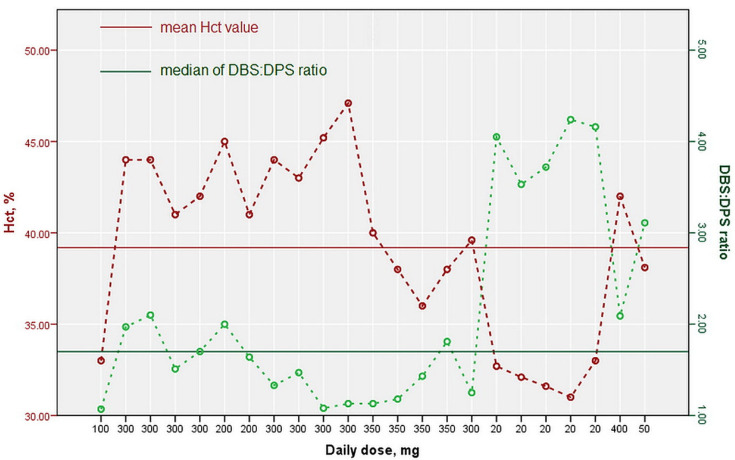
Diagram of ZNS erythrocyte to plasma ratio regarding the Hct % and daily dose of ZNS.

**Table 1 molecules-27-04899-t001:** MRM transitions and corresponding MS/MS tune parameter.

Analyte	Molecular Ion, *m*/*z*	Production, *m*/*z*	Collision Energy (V)	Cone Voltage (V)	Capillary Voltage (kV)
Zonisamide	212.86	131.85 *	16	30	3.8
		76.95	30	30	3.8
(1-(2,3-dichlorphenyl) piperazine)-IS	231.19	153.01 *	22	42	3.8
		117.15	46	42	3.8

* The most abundant transition used for quantification.

**Table 2 molecules-27-04899-t002:** Important parameters of linear weighted regression models.

**DPS**					
**Linear Regression Models**		**Weighting Factor**	**r**	**The Largest%RE * for Non-LLOQ Concentrations, %**	**% RE * for the LLOQ Concentration, %**
** *a* **	** *b* **				
0.20506	0.03649	1/x^2^	0.994	14.3%	2.13%
0.18083	0.03579	1/x^2^	0.994	13.3%	0.90%
0.19246	0.02097	1/x^2^	0.993	14.3%	3.79%
0.15495	0.03605	1/x^2^	0.998	6.73%	2.84%
DBS, Hct 42%					
**Linear Regression Models**		**Weighting Factor**	**r**	**The Largest%RE * for Non-LLOQ Concentrations, %**	**% RE * for the LLOQ Concentration, %**
** *a* **	** *b* **				
0.12987	0.00829	1/x^2^	0.991	13.7	0.09
0.25000	−0.00509	1/x^2^	0.9990	6.24	3.04
0.25522	0.02071	1/x^2^	0.992	12.9	3.64
0.34892	0.016996	1/x^2^	0.995	14.4	7.51

*a*—slope; *b*—intercept; r—coefficient of correlation; * % RE—percentage relative error, calculated as ǀ(calculated—nominal concentration)/nominal concentrationǀ × 100, represents deviation of the calculated from the nominal concentration corrected for the nominal concentration and expressed as a percentage value.

**Table 3 molecules-27-04899-t003:** Accuracy and precision testing for the DPS and DBS samples.

	Accuracy And Precision Testing									
**Within-run accuracy and precision**		**Dried Plasma Samples (DPS)**					**Dried Blood Samples (DBS)**			
	Analytical run		**LLOQ** **0.250 µg/mL**	**Low QC1** **0.750 µg/mL**	**Medium QC2** **7.5 µg/mL**	**High QC3** **38** **µg/mL**	**LLOQ** **0.125 µg/mL**	**Low QC1** **0.375 µg/mL**	**Medium QC2** **3.75 µg/mL**	**High QC3** **19 µg/mL**
	**I**	<C_det._>, µg/mL	0.242	0.693	6.67	33.45	0.134	0.411	4.08	19.71
		**<bias>,%**	**−3.21**	**−7.55**	**−11.11**	**−11.98**	**6.90**	**9.60**	**8.74**	**3.76**
		SD,µg/mL	0.021	0.051	0.17	0.42	0.011	0.021	0.158	1.85
		**RSD,%**	**8.64**	**7.40**	**2.62**	**1.26**	**8.41**	**5.18**	**3.87**	**9.39**
	**II**	<C_det._>, µg/mL	0.248	0.707	7.11	33.72	0.127	0.361	3.77	19.76
		**<bias>,%**	**−0.86**	**−5.76**	**−5.19**	**−11.26**	**1.87**	**3.76**	**0.60**	**3.98**
		SD,µg/mL	0.039	0.065	0.46	0.62	0.020	0.032	0.461	1.5034
		**RSD,%**	**15.85**	**9.25**	**6.45**	**1.85**	**15.37**	**8.82**	**12.23**	**7.61**
	**III**	<C_det._>, µg/mL	0.246	0.779	7.64	35.14	0.117	0.391	4.12	19.92
		**<bias>,%**	**−1.62**	**3.92**	**1.85**	**0.40**	**−6.67**	**4.34**	**9.82**	**4.85**
		SD,µg/mL	0.037	0.040	0.36	1.67	0.012	0.045	0.212	2.014
		**RSD,%**	**14.98**	**5.15**	**4.75**	**4.75**	**9.92**	**11.53**	**5.15**	**10.11**
**Between-run accuracy and precision**			**LLOQ** **0.250 µg/mL**	**Low QC1** **0.750 µg/mL**	**Medium QC2** **7.5 µg/mL**	**High QC3** **38 µg/mL**	**LLOQ** **0.125 µg/mL**	**Low QC1** **0.375 µg/mL**	**Medium QC2** **3.75 µg/mL**	**High QC3** **19 µg/mL**
		<C_det._>, µg/mL	0.245	0.727	7.14	34.10	0.126	0.388	3.99	19.80
		**<bias>,%**	**−1.90**	**−3.13**	**−4.81**	**−10.26**	**0.70**	**3.39**	**6.39**	**4.20**
		SD,µg/mL	0.031	0.063	0.52	1.24	0.015	0.038	0.326	1.671
		**RSD,%**	**12.63**	**8.67**	**7.35**	**3.65**	**12.21**	**9.84**	**8.17**	**8.44**
		n	15	15	15	15	15	15	15	15

<Cdet.>—average determined ZNS concentration calculated by taking into account 5 replications per each concentration level for within-run accuracy and precision evaluation, i.e., 15 replications per each concentration level for between-run accuracy and precision evaluation; <bias>—average percentage bias represents the deviation of average ZNS concentration determined at certain concentration level form certain nominal concentration—((<Cdet.>-nominal concentration)/nominal concentration) × 100; SD—standard deviation calculated for ZNS concentration determined at certain concentration level; RSD—percentage coefficient of variation for ZNS concentrations determined at certain concentration levels.

**Table 4 molecules-27-04899-t004:** Matrix effect estimated via matrix factors of ZNS and IS and IS normalized matrix factor.

**DBS**								
	MF_ZNS_ * %	MF_IS_ * %	MF_ZNS_/MF_IS_ %	MP <Cc> * µg/mL	MP C_ZNS_ RSD, %	MA <Cc> µg/mL	MA C_ZNS_ RSD, %	C_ZNS_ bias, %
low QC 0.375 µg/mL	110.11	108.26	101.72	0.417	3.73	0.409	4.33	2.85
medium QC 3.75 µg/mL	91.55	91.02	100.58	4.19	2.51	4.15	6.61	0.90
high QC 19 µg/mL	99.12	95.41	103.89	21.19	4.24	20.63	5.31	2.70
n	3							
**DPS**								
	MF_ZNS_ * %	MF_IS_ * %	MF_ZNS_/MF_IS_ %	MP <Cc> * µg/mL	MP C_ZNS_ RSD, %	MA <Cc> µg/mL	MA C_ZNS_ RSD, %	C_ZNS_ bias, %
low QC 0.750 µg/mL	102.26	99.44	102.84	0.832	2.25	0.804	7.27	3.58
medium QC 7.5 µg/mL	93.92	96.62	97.21	8.17	4.77	8.41	1.16	−2.84
high QC 38 µg/mL	100.11	102.30	97.86	40.91	6.14	41.91	3.09	−2.39
n	3							

MF_ZNS_—Matrix factor of ZNS calculated by dividing average peak area of ZNS in matrix-present sample (MP P_zns_) with average peak area of ZNS in matrix-absent sample (MA P_zns_), then multiplied by 100 to express as percentage value—MP P_zns_/MA P_zns_ × 100; MFIS—Matrix factor of IS calculated by dividing average peak area of IS in matrix-present sample (MP P_is_) with average peak area of IS in matrix-absent sample (MA Pis), then multiplied by 100 to express as percentage value—MP Pis/MA Pis ×100; MF_ZNS_/MF_IS_—IS normalised MF, calculated by dividing the MF of the ZNS by the MF of the IS and multiplying by 100 to express as percentage value; MP <Cc>—average ZNS concentration calculated for matrix-present samples; MA <Cc>—average ZNS concentration calculated for matrix-absent samples; MP C_ZNS_ RSD,%—percentage coefficient of variation for ZNS concentration calculated at certain QC level for both batches of matrix in matrix-present samples; MA C_ZNS_ RSD,%—percentage coefficient of variation for ZNS concentration calculated at certain QC level in matrix-absent samples; C_ZNS_ bias%-percentage deviation of ZNS concentration determined in the matrix-present samples relative to ZNS concentration determined in matrix-absence sample, calculated by dividing deviation of the average ZNS concentration obtained in matrix-present samples from average ZNS concentration obtained in matrix-absent samples, with ZNS determined in matrix-absent samples—((MP <Cc> − MA <Cc>)/MA <Cc>) × 100; n—number of replicates per each QC level; * average values calculated by considering all individual values at each QC level obtained from the both batches of each matrix in the case of the matrix-present samples and all individual values at each QC level for the matrix-absent samples.

**Table 5 molecules-27-04899-t005:** Extraction Recovery evaluation.

**DBS**			
ZNS	<R_ZNS_>, %	IS	<R_IS_>, %
low QC 0.375 µg/mL	103.24	20 ng/mL	101.11
medium QC 3.75 µg/mL	101.19		107.16
high QC 19 µg/mL	100.83		102.30
overall <R>, %	101.75	overall <R>, %	103.52
SD	1.30	SD	3.21
RSD, %	1.28	RSD, %	3.10
**DPS**			
ZNS	<R_ZNS_>, %	IS	<R_IS_>, %
low QC 0.750 µg/mL	101.62	20 ng/mL	103.19
medium QC 7.5 µg/mL	95.83		99.82
high QC 38 µg/mL	94.32		99.08
overall <R>, %	97.26	overall <R>, %	100.70
SD	3.58	SD	2.19
RSD, %	3.96	RSD, %	2.18

n = 3, where n represent number of replication per each QC level; <R_ZNS_>—average value of ZNS extraction recovery, calculated by dividing average ZNS peak area from DMS (P_ZNS_) with average ZNS peak area from post-extraction added working solution of ZNS (P_ZNSpe_); P_ZNS_/P_ZNSpe_ × 100; <R_IS_>—average value of IS extraction recovery, calculated by dividing average IS peak area from DMS with average IS peak area from post-extraction spiked samples; overall <R>—average value of recovery calculated basing to the average Recovery value obtained for each QC level; SD—standard deviation of average Recovery values calculated for all QC levels; RSD—a coefficient of variation that represent the variation in average Recovery value of ZNS and IS calculated for all QC levels; The results indicated consistent and reproducible Recovery values, demonstrating the uniform contribution of the extraction procedure to ZNS quantification regardless the change in concentration, i.e., the extraction procedure was efficient and reproducible.

**Table 6 molecules-27-04899-t006:** Effects of Hct on the reliability (accuracy and precision) of the ZNS quantification in DBS.

Hct,%		Linear Regression Model					
		*a*		*b*		w.f.	r
20.2		0.25612		0.01996		1/x^2^	0.9930
30.0		0.23202		0.02940			0.9893
42.0		0.25522		0.02071			0.9923
50.2		0.28783		0.01758			0.9933
60.0		0.26068		0.01584			0.9938
**Accuracy and Precision**							
Conc. level	nominal concentration, µg/mL	Hct,%	bias set 1,%	bias set 2,%	RSD set 1,%	RSD set 2,%	bias set1/set2,%
LLOQ	0.125	20.2	13.96	11.99	2.97	3.03	1.76
QC1	0.375		1.15	0.71	2.86	2.88	0.43
QC2	3.75		8.27	8.57	3.06	3.06	−0.28
QC3	19.0		10.58	10.95	2.58	2.58	−0.34
LLOQ	0.125	30.0	−9.45	9.55	4.86	3.65	−17.34
QC1	0.375		0.71	0.63	6.70	6.10	0.08
QC2	3.75		9.86	0.78	2.05	2.03	9.01
QC3	19.0		6.31	−3.17	7.11	7.09	9.79
LLOQ	0.125	42.0	−6.67	−6.67	9.92		0
QC1	0.375		4.34	4.34	11.53		0
QC2	3.75		9.82	9.82	5.15		0
QC3	19		4.85	4.85	10.11		0
LLOQ	0.125	50.2	−4.45	−2.06	5.53	6.08	−2.44
QC1	0.375		−0.07	9.42	5.73	5.90	−8.68
QC2	3.75		−7.89	3.55	8.49	8.52	−11.05
QC3	19.0		−2.78	9.58	2.79	2.79	−11.28
LLOQ	0.125	60.0	6.64	−6.37	6.63	7.71	13.90
QC1	0375		0.27	−2.68	7.38	7.77	3.03
QC2	3.75		1.88	3.55	8.48	8.52	−1.61
QC3	19.0		8.95	11.18	4.46	4.46	−2.01
n	5						

*a*—slope; *b*—intercept; w.f.—weighting factor; r—coefficient of correlation; bias set 1—%bias of calculated average concentration for the first set and nominal concentration (calculated average concentration– nominal QC concentration)/nominal QC concentration) × 100); concentration calculated by application of regression models derived for the calibration standards with appropriate Hct. bias set 2—%bias of calculated average concentration for the second set and nominal concentration (calculated average concentration– nominal QC concentration)/nominal QC concentration) × 100); concentration calculated by application of regression models derived for the calibration standards with nominal Hct of 42%; RSD set 1—percentage coefficient of variation calculated at certain concentration level for the first set of concentration; RSD set 2—percentage coefficient of variation calculated at certain concentration level for the second set of concentration; N—number of observations per each concentration level.

**Table 7 molecules-27-04899-t007:** Results of the analysis of the patients’ DBS and DPS samples.

Patient’ Sample Label	Hct,%	Daily Dose, mg	DBS Concentration, µg/mL	DPS Concentration, µg/mL	DBS:DPS Ratio
1	33.0	100	30.45	28.55	1.07
2	44.0	300	17.89	9.10	1.97
3	44.0	300	16.26	7.75	2.10
4	41.0	300	23.69	15.64	1.51
5	42.0	300	15.01	8.81	1.70
6	45.0	200	16.90	8.44	2.00
7	41.0	200	13.43	8.18	1.64
8_1	44.0	300	18.72	14.08	1.33
8_2	43.0		15.93	10.83	1.47
8_3	45.2		13.42	12.46	1.08
8_4	47.1		17.42	15.45	1.13
9_1	40.0	350	26.39	23.31	1.13
9_2	38.0		27.38	23.14	1.18
9_3	36.0		26.50	18.55	1.43
9_4	38.0		27.84	15.36	1.81
10	39.6	300	29.62	23.74	1.25
11_0	32.7	20, 40 *	3.83	0.95	4.05
11_1	32.1		4.41	1.25	3.53
11_2	31.6		5.52	1.48	3.72
11_3	31.0		5.55	1.31	4.24
11_4	33.0		6.01	1.45	4.16
12	42.0	400	79.70	38.04	2.09
13	38.1	50	16.66	5.36	3.11
**Overall Descriptive Statistic of Conducted Analysis**					
	**Hct,%**	**Daily Dose, mg**	**DBS Concentration,** **µg/mL**	**DPS Concentration,** **µg/mL**	**DBS:DPS Ratio**
mean ± SD **	39.19 ± 4.99	/	/	/	/
range	31–47.1	20–400	3.83–79.70	0.95–38.04	1.07–4.24
median	/	300	16.90	10.83	1.70
interquartile range	/	150 i.e., 150–300	13.08 i.e., 13.42–26.50	13.19 i.e., 0.95–38.04	1.86 i.e., 1.25–3.11
n	23	13	23	23	23

* Ten days before sampling ZNS included in therapy in daily dose of 20 mg; five days before sampling daily dose was changed from 20 mg to 40 mg; ** a mean value accompanied with standard deviation (SD) displayed only for the data with normal distribution; normality of distribution was evaluated by Shapiro–Wilk test of normality in SPSS 20 software; n—number of observation.

## Data Availability

Data is contained within the article.

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
