# Peer review of "Quantification of Zonisamide in Dried Blood Spots and Dried Plasma Spots by UPLC–MS/MS: Application to Clinical Practice"

_molecules, 2022, doi:10.3390/molecules27154899_

Round 1

Reviewer 1 Report

In this study Authors have described a UHPLC-MS/MS method developed and validated for the determination of zonisamide in dried plasma spots (DPS) and dried blood spots (DBS). Authors have followed all FDA guidelines for Method development and validation.

In order to recommend this study for editorial decision Authors need to clarify following points:

1) LC-MS/MS methods for quantification of Zonisamide have been described earlier.

(doi: 10.1016/j.jchromb.2014.05.017) 

(doi: 10.1097/FTD.0000000000000516)

Except using DBS as starting matrix what is unique about this study. Authors need to clarify this and include more details in introduction about not using already established methods.

2) Authors need to describe the reason of using  1-(2,3-dichlorphenyl)piperazine, as an Internal standard in this study instead of labelled IS.

Author Response

Reviewer #1: correction highlighted in green

  1. In this study Authors have described aUHPLC-MS/MS method developed and validated for the determination of zonisamide in dried plasma spots (DPS) and dried blood spots (DBS). Authors have followed all FDA guidelines for Method development and validation.

In order to recommend this study for editorial decision Authors need to clarify following points:

  1. LC-MS/MS methods for quantification of Zonisamide have been described earlier.

(doi: 10.1016/j.jchromb.2014.05.017) 

(doi: 10.1097/FTD.0000000000000516)

Except using DBS as starting matrix what is unique about this study. Authors need to clarify this and include more details in introduction about not using already established methods.

The authors thank the reviewer for this valuable comment. The appropriate amendments were introduced in the manuscript - two suggested publications and critical analysis of previously reported studies.

We kindly point out that all the previously published LC-MS/MS or UHPLC-MS/MS methods were developed, validated and applied for the determination of zonisamide in plasma or serum and as such were not appropriate for implementation in our study.

So far, method suitable for quantification of zonisamide in both plasma and blood was not proposed, providing means to correlate obtained concentrations and apply findings in routine clinical practice. Therefore, the aim of this research was to develop the first UPLC–MS/MS method for the quantification of zonisamide in DPSs and DBSs with the aid of image processing for precise estimation of DBS and DPS areas after scanning of the Guthrie card on which the appropriate matrix volume was spotted.

Further more, DBS-based method was validated using 5 Hct levels (20.2%, 30%, 42%, 50.2% and 60%) to determine the range within which the method is independent of Hct or dependency is within acceptable limits.

  1. Authors need to describe the reason of using 1-(2,3-dichlorphenyl)piperazine, as an Internal standard in this study instead of labeled IS.

We thank the reviewer for this question. We are aware that nowadays, a stable isotope labeled internal standards (SIL IS), are preferred for a quantification using LC/MS/MS assays. However, considering some demerits which could be expected during the utilization of the SIL IS (especially deuterium labeled), high price of solely available deuterium labeled zonisamide (ZNS-D4) and our limited financial abilities, we decided to use structurally analogue IS. The selection of the 1-(2,3-dichlorphenyl)piperazine as the IS is supported with the following:

  • selected IS and ZNS have aligned physical-chemical properties, primarily hydrophobicity and ionization characteristic/behavior (ionized in positive ionization mode),
  • selected IS does not correspond to any approved drug, which potentially could be co-administered, as well as any in vivo metabolic product,
  • during the validation, as part of selectivity testing it was confirmed that IS does not contribute, i.e. interfere to the ZNS signal.

Reviewer 2 Report

The work introduced is outstanding, just some points for improvement:

-        In introduction you mention: (For the determination of zonisamide in plasma/serum HPLC [3, 4], LC/MS [5], and 45 LC–MS/MS [6] methods were reported, while in dried plasma spots (DPSs) it was quantified by HPLC-UV method [7]).  Accordingly, at end of discussion, it may be suitable to highlight advantages of newly introduced method against these published ones considering figures of merit.

-        In methodology, you should mention how the optimum analysis conditions were reached and why? (method optimization).

-        It is trending now to assess greenness of your newly introduced method, so it would be recommended to use one of the tools of assessment (I highly recommend AGREE tool which provides both qualitative and quantitative measures). This paper will help you (https://doi.org/10.3390/separations9060147 )

Author Response

Reviewer #2: correction highlighted in yellow

The work introduced is outstanding, just some points for improvement:

  1. In introduction you mention: (For the determination of zonisamide in plasma/serum HPLC [3, 4], LC/MS [5], and LC–MS/MS [6] methods were reported, while in dried plasma spots (DPSs) it was quantified by HPLC-UV method [7]). Accordingly, at end of discussion, it may be suitable to highlight advantages of newly introduced method against these published ones considering figures of merit.

The authors thank reviewer for this comment. The advantages of our method against previously published are highlighted at the end of Conclusion.

  1. In methodology, you should mention how the optimum analysis conditions were reached and why? (method optimization).

During preliminary experiments (screening phase) it was observed that in the whole experimental domain investigated factors (acetonitrile content from 80 - 90%, formic acid content from 0.05 - 0.1%, as well as column temperature from 25 - 40°C) were in agreement and with a positive effect on ZNS signal and retention. Optimal mobile phase conditions were defined quite straightforward. The appropriate explanation added to the text.

  1. It is trending now to assess greenness of your newly introduced method, so it would be recommended to use one of the tools of assessment (I highly recommend AGREE tool which provides both qualitative and quantitative measures). This paper will help you (https://doi.org/10.3390/separations9060147)

We thank to the reviewer for drawing our attention to this very interesting research dealing with assessment of the environmental impacts of the analytical/chromatographic methods, in a simple, straightforward way. The greenness of the method was evaluated in the proposed way, and the results of evaluation incorporated into the manuscript as 3.4. Chapter. The Analytical greenness report with graphical presentation of the assessment is pro-vided as Supplementary figure 1 (Figure S1).

Round 2

Reviewer 1 Report

Authors have addressed all my comments. I recommend this study for publication.